# Electronic cigarettes as a smoking cessation aid for patients with cancer: beliefs and behaviours of clinicians in the UK

Jo Brett ,[1] Emma L Davies,[2] Fiona Matley,[1] Paul Aveyard ,[3] Mary Wells ,[4] David Foxcroft ,[2] Brian Nicholson ,[3] Shiroma De Silva Minor,[5] Lesley Sinclair,[6] Sarah Jakes,[7] Eila Watson [1]

For numbered affiliations see end of article.

**Correspondence to**
Dr Jo Brett;
jbrett@brookes.ac.uk

## ABSTRACT

**Objectives** To explore UK clinicians' beliefs and behaviours around recommending e-cigarettes as a smoking cessation aid for patients with cancer.

**Design** Cross-sectional online survey.

**Setting** England, Wales, Scotland and Northern Ireland.

**Participants** Clinicians involved in the care of patients with cancer.

**Primary and secondary outcomes** Behavioural Change Wheel capability, opportunity and motivation to perform a behaviour, knowledge, beliefs, current practice around e-cigarettes and other smoking cessation practices.

**Method** Clinicians (n=506) completed an online survey to assess beliefs and behaviours around e-cigarettes and other smoking cessation practices for patients with cancer. Behavioural factors associated with recommending e-cigarettes in practice were assessed.

**Results** 29% of clinicians would not recommend e-cigarettes to patients with cancer who continue to smoke. Factors associated with recommendation include smoking cessation knowledge (OR 1.56, 95% CI 1.01 to 2.44) and e-cigarette knowledge (OR 1.64, 95% CI 1.06 to 2.55), engagement with patients regarding smoking cessation (OR 2.12, 95% CI 1.12 to 4.03), belief in the effectiveness of e-cigarettes (OR 2.36 95% CI 1.61 to 3.47) and belief in sufficient evidence on e-cigarettes (OR 2.08 95% CI 1.10 to 4.00) and how comfortable they felt discussing e-cigarettes with patients (OR 1.57 95% CI 1.04 to 2.36).

**Conclusion** Many clinicians providing cancer care to patients who smoke do not recommend e-cigarettes as a smoking cessation aid and were unaware of national guidance supporting recommendation of e-cigarettes as a smoking cessation aid.

## BACKGROUND

Smoking is a well-established risk factor for many common cancers.[1–3] The adverse effects of smoking continue after a cancer diagnosis, increasing the risk of treatment-related complications, recurrence, the development of a second primary cancer and mortality from both cancer-related and non-cancer-related causes.[4–11] Despite the increased risk

### Strengths and limitations of this study

► This study reports an online survey with a wide-range of clinicians to assess their beliefs and current practices around e-cigarettes as a smoking cessation aid for cancer patients.

► The survey used the capability, opportunity and motivation (COM-B) behavioural model to understand the factors that influence clinicians to recommend or not recommend e-cigarettes.

► Quota sampling enabled representation across relevant clinical roles in primary and secondary care and across geographical areas in the UK.

► The sample is limited to UK clinicians where public health policy supports use of e-cigarettes as a smoking cessation aid for risk reduction compared to tobacco cigarette smoking.

► The sampling was not random and so participation could have been affected by whether clinicians were interested in the topic, although paying clinicians to complete the study aimed to mitigate this.

of complications of cancer treatment, recurrence and death, many patients with smoking-related cancers continue smoking following diagnosis[12–15] having tried and failed to stop smoking. Effective aids for cessation are available, but support to quit is not routinely offered as part of cancer care.[16 17] One study found that 39% (n=1129) of patients with lung cancer, 37% (n=281) upper aerodigestive tract cancer patients and 49% (n=850) of patients with bladder cancer continued to smoke 1 year after diagnosis, figures that were likely to be higher as a third of the potential participants' smoking status was unknown.[18] To enhance the length and health-related quality of their lives, efforts are needed to support cancer patients to stop smoking cigarettes.

In recent years, e-cigarettes have grown in popularity as a cessation aid among smokers

worldwide, but the e-cigarette regulatory environment in each country varies considerably due to policy-makers' reactions to a rapidly developing evidence base.[19 20] Often this is influenced by strategic positions on tobacco control, for example, harm prevention versus harm reduction,[21] and whether e-cigarettes are classified as a tobacco, medicinal or consumer product.[19] The main debates focus on trying to achieve a balance between the risks and potential of e-cigarettes.[21]

Support for the use of e-cigarettes for smoking cessation is endorsed by several health-related organisations in the UK including Public Health England,[22] Cancer Research UK,[23] the National Health Service (NHS)[24] and the National Centre for Smoking Cessation and Training.[25] Public Health Scotland,[26] Public Health Wales,[27] the British Medical Association,[28] the Royal College of General Practitioners[29] and the Royal College of Physicians[30] also acknowledge that e-cigarettes are considered less harmful than smoking tobacco cigarettes or that some people may find using e-cigarettes useful for stopping or reducing smoking.

In the UK and other European countries, the manufacture, presentation and sales of e-cigarettes are regulated by the Tobacco Products Directive (2014/40/EU).[31 32] The regulation prohibits sales of e-cigarettes to people under 18 years, most forms of advertising and places restrictions on the type and quality of ingredients. In France, there are further restrictions on the use of e-cigarettes in public places.[33] The regulatory environment differs substantially in other parts of the world.

In the USA, for instance, the Centres for Disease Control and Prevention advice is not unlike many UK public health bodies in stating that 'if (adults) choose to use e-cigarettes as an alternative to cigarettes, they should completely switch from cigarettes to e-cigarettes and not partake in an extended period of dual use of both products that delays quitting smoking completely'.[34] The WHO position is that there is insufficient evidence concerning comparisons with combustible cigarettes or efficacy for smoking cessation, and that the use of e-cigarettes is harmful,[35] although In October 2018, 72 experts with no connections to the tobacco industry wrote to the WHO Director-General to argue that WHO should embrace innovation and more actively include tobacco harm reduction in its strategy to tackle the burden of smoking-related disease.

While the need for further evidence around impacts of e-cigarettes on cessation and long-term harms is widely acknowledged, many health-related organisations also recognise that e-cigarettes may have a potential role in smoking cessation and could help people that may otherwise continue to smoke. A risk reduction policy to encourage smoking cessation is particularly important in people who have been diagnosed with cancer who continue smoking and therefore increase their risk of recurrence, other comorbidities and premature death as a result.[4-11]

In the UK, e-cigarettes have swiftly become the most popular smoking cessation product for smokers.[36 37]

There are 3.6 million e-cigarettes users in the UK, of which only 0.8% have never smoked.[37] E-cigarettes are used in 30% of quit attempts with currently around 20% of smokers and 30% of recent ex-smokers using them.[38] Evidence shows that e-cigarettes help smokers to stop smoking long term[39-41] and a recent study suggests that e-cigarettes are more effective for smoking cessation than nicotine-replacement therapy, when both products are accompanied by behavioural support.[42] However, qualitative evidence implies that clinicians may be uncertain about the use of e-cigarettes as a cessation aid, with some expressing hostility and reporting practices that are not consonant with the evidence.[43]

The aims of this study were to understand clinicians' beliefs and behaviours related to e-cigarettes for patients with cancer who continue to smoke, and to understand the behavioural factors that may promote or inhibit recommending e-cigarettes.

## METHODS
### Design
The study was a cross-sectional online survey of clinician's knowledge, beliefs and current practice of smoking cessation and e-cigarettes. The survey was sent to clinicians involved in the adult cancer care pathway, working in primary and secondary care.

### Survey development
The survey was developed using methods suggested by Bowling.[44] The questions drew on (1) a literature review to identify evidence of clinicians' knowledge, attitudes, behaviours and current practice with respect to smoking cessation interventions, including e-cigarettes, in patients with cancer; (2) expert opinion; (3) drawing on Behaviour Change Wheel (BCW).[45 46] Development also drew on previous survey questions exploring attitudes to e-cigarettes.[47-49] The face validity of the survey was evaluated with five general practitioners (GPs) and five cancer clinicians to evaluate whether the survey was appropriate and the questions were understood. This resulted in minor modification of wording of some questions.

The final questionnaire was structured according to the COM-B behaviour model based on Michie's BCW for development of interventions.[45 46] This model proposes that people need capability (C), opportunity (O) and motivation (M) to perform a behaviour (B) and was developed to guide understanding of behaviour in context and develop behavioural targets. The model proposes that for someone to engage in a particular behaviour at a given moment they must be physically able and have the psychological ability and have the social and physical opportunity to enact the behaviour and, in addition, want or need to enact the behaviour more than any other competing behaviours at that moment. This inclusive definition of motivation covers basic drives and automatic processes such as habit and impulses as well as reflective processes including intention and choice. If a desired behaviour is

not occurring (or an undesirable behaviour occurring) then an analysis of the determinants of the behaviour will help to define what needs to shift in order for the desired behaviour to occur (or the unwanted behaviour to cease).

The questionnaire included items relating to psychological capability (knowledge of smoking status of patients, knowledge and skills about smoking cessation in general, knowledge and skills about of e-cigarettes); physical opportunity (time constraints talking to patients about e-cigarettes, physical constraints due to policy); social opportunity (relationship with patient and how this impacts on smoking cessation advice given, social norms, for example, most my colleagues support use of e-cigarettes, most my colleagues feel uncomfortable recommending e-cigarettes, clinicians should discourage smoking); reflective motivation (motivation to engage in smoking cessation with cancer patients, beliefs about how effective e-cigarettes are for patients with cancer, beliefs about the clinician's role, beliefs about the evidence base around e-cigarettes, attitudes towards e-cigarettes, beliefs about the harms of e-cigarettes particularly in comparison to tobacco cigarettes and automatic motivation (how comfortable they feel giving smoking cessation advice in general and in giving specific advice on e-cigarettes to patients). A full copy of the questionnaire is available electronically from the authors.

### Sample

The survey was distributed electronically by M3, the research arm of doctors.net.uk, a leading market research consultancy specialising in high-quality online research using pre-recruited panels of medical professionals. All clinicians who are registered with doctors.net.uk have to provide their clinical registration number during the registration process. M3 have 7781 GPs, 436 oncologists, 708 surgeons, 221 cancer nurse specialists (CNSs) and 315 practice nurses on their research panels. Sampling was restricted to currently practising clinicians, was stratified by NHS region, and was conducted on a 'first come, first served' basis; the target number of responses was 100 for each of the clinician types. When the quota of responses from each type of clinician or from each UK NHS region was reached, the survey was closed for that clinician group or region.

The survey was completed between November 2018 and February 2019. A small financial incentive was offered. Participants confirmed they had read the participant information and consented to take part via email before the survey link was sent.

### Analyses

Anonymised electronic responses were imported into SPSS (V.25) for analysis. Frequencies and proportions were used to summarise questionnaire responses. Results are reported using the COM-B model: physical and psychological capability; physical and social opportunity; and automatic and reflective motivation.

Means and SD for each measure for the whole sample were calculated and then compared by clinician occupation. One-way analysis of variance was used to compare means for the other measures. Multiple comparisons were taken into account using a Bonferroni correction. Student-Newman-Keuls post hoc tests were used to examine differences reported.

$X^2$ tests were used to compare the clinicians on categorical measures. Binomial logistic regression models were used to predict the likelihood of recommending e-cigarettes to cancer patients. The dependent variable was dichotomised by always/nearly always/often recommend e-cigarettes (Q5_3) vs sometimes/infrequently/never.

### Patient and public involvement

Cancer service users and vapour representatives were involved in the proposal development, questionnaire development and dissemination of the results of this study. One vapour representative was involved in the write up and is an author on the paper.

## RESULTS

### Characteristics of respondents

The online survey was completed by 506 clinicians: 103 GPs, 102 oncologists, 100 cancer surgeons, 102 practice nurses and 99 CNSs. One CNS was excluded because they were a CNS for children. Table 1 describes the clinicians' characteristics.

### Reported behaviour

Twenty-nine precent n=147) of clinicians would not recommend e-cigarettes to patients with cancer who smoke. Just over half (51%, n=258) would recommend e-cigarettes as an interim measure, to help patients stop smoking completely, while 20% (n=101) would recommend e cigarettes as a partial replacement for smoking tobacco.

### Psychological capability

Most clinicians (78%, n=394) knew the smoking status of their patients with cancer, and routinely recorded their smoking status (73%, n=368). Sixty-seven per cent (n=339) reported that they routinely recommended patients stop smoking, or cut down (52%, n=263). Twenty-nine per cent (n=147) referred patients to the NHS stop smoking services, 14% (n=71) recommended nicotine replacement therapy, 9% (n=46) recommended digital smoking cessation tools and 5% (n=25) prescribed medication (varenicline or bupropion).

Many clinicians felt they had insufficient knowledge (57%, n=286) and training (73%, n=370) to provide advice about e-cigarettes to patients and a further 36% (n=182) indicated that they did not know the efficacy of e-cigarettes with regard to smoking cessation.

Clinicians derived information about e-cigarettes from several sources. Overall, only 9.5% (n=48) of clinicians knew whether their organisation had guidance

**Table 1** Clinician characteristics

| Demographics | All participants % (n=506) | Primary care % (n=205) | Secondary care % (n=301) |
|---|---|---|---|
| Gender | | | |
| Male | 41.1 | 35.1 | 45.2 |
| Female | 57.5 | 63.9 | 53.2 |
| Prefer not to say | 1.4 | 1.0 | 1.7 |
| Years of professional experience | 1995 (1968–2009) | 1994 | 1996 |
| Role | | | |
| General practitioner | 20.4 | 40.5 | |
| Practice nurse | 20.2 | | |
| Cancer surgeon | 19.8 | | 59.5 |
| Oncologist | 20.2 | | |
| Cancer nurse specialist | 19.6 | | |
| NHS region | | | |
| London | 15.8 | 12.2 | 18.3 |
| South of England | 21.1 | 21.5 | 20.9 |
| Midlands and East SEA | 25.5 | 25.4 | 25.6 |
| North of England | 20.6 | 22.9 | 18.9 |
| Scotland | 12.1 | 13.7 | 11.0 |
| Wales | 2.2 | 2.4 | 2.0 |
| Northern Ireland | 2.8 | 2.0 | 3.3 |
| Main cancer group they care for | | | |
| All | 37.0 | 76.1 | 10.3 |
| Breast | 24.1 | 12.2 | 32.2 |
| Prostate | 19.4 | 13.7 | 23.3 |
| Lung/mesothelioma | 18.0 | 11.7 | 22.3 |
| Bowel | 11.9 | 5.4 | 16.3 |
| Kidney | 11.9 | 0.5 | 19.6 |
| Bladder | 11.3 | 0.5 | 18.6 |
| Other cancer groups | <10 | <5 | <15 |
| Smoking status of clinicians | | | |
| Smoke tobacco cigarettes | 2.0 | 1.5 | 2.3 |
| Use e-cigarettes** | 1.2 | 1.5 | 1.0 |
| Ex-smoker | 21.3 | 22.9 | 20.3 |
| Never smoker | 72.3 | 72.2 | 72.4 |
| Other | 1.4 | 1.5 | 1.3 |
| Prefer not to say | 3.6 | 2.4 | 4.3 |

NHS, National Health Service.

concerning advice to patients on e-cigarette use. Most clinicians had sought information about e-cigarettes from government/health agencies (55%), but also from professional associations (37%), healthcare colleagues (29%), news/media/advertising (24%), scientific literature (23%), professional development/training (22%) and charities (18%). Nineteen per cent of clinicians had never sought information about e-cigarettes. One-quarter of respondents (25%, n=124) were uncertain whether e-cigarettes were less harmful than smoking tobacco, while 10% (n=52) thought e-cigarettes were equally harmful or more harmful than smoking tobacco. Eighteen per cent (n=93) considered using e-cigarettes to be more harmful than regular nicotine replacement therapies (eg, gum, nasal spray, patches) and 54% (n=273) were uncertain.

## Physical opportunity

The majority of clinicians (56%, n=285) said that e-cigarette use was prohibited in all areas at their main place of work. Twenty-three per cent (n=119) reported that the use of e-cigarettes was permitted, with 24% (n=29) reporting use in designated smoking areas only. Overall, 51% (n=258) agreed that time constrained their ability to talk about e-cigarettes with patients.

## Social opportunity

The nature of the relationships between clinicians and patients with cancer were important in whether and how smoking cessation was discussed. Fifty-five per cent (n=278) of clinicians reported that having a good relationship would make them more likely to speak to their patient about stopping or cutting down, while 45% (n=228) reported that having a poor relationship would make them less likely to discuss smoking cessation. Many clinicians (42% n=212) felt uncomfortable when asked by patients for an opinion on e-cigarettes. The large majority, 82%, had been asked about e-cigarettes by patients in the past year (2017/18), up from 21% in 2016/2017.

Thirty-eight per cent (n=192) said that most of their colleagues would feel uncomfortable recommending e-cigarettes to patients with cancer, and 37% (n=187) were unsure whether clinicians should discourage patients with cancer from using e-cigarettes.

## Automatic motivation

Subconscious biases towards e-cigarettes were influenced by clinicians' beliefs around the effectiveness of and evidence on e-cigarettes, as reported in the psychological capability section above.

Clinicians reported that their decisions to speak with patients with cancer about smoking cessation were influenced by their perceptions of the patient. Clinicians reported that they were more likely to discuss smoking cessation if they judged the person was motivated to quit (69%, n=349), or was coping well (67%, n=339).

## Reflective motivation

Reflective motivation related to the clinicians' perceived role in smoking cessation and national and organisational policy on e-cigarettes. Two-thirds (65%, n=327) of clinicians agreed that they should play a greater role in helping cancer patients stop smoking. Clinicians were divided over whether e-cigarettes should be licensed and available on prescription for patients with cancer, with 39% (n=199) respondents disagreeing, and 32% (n=162) saying they should be available on prescription. Furthermore, 30% (n=150) of clinicians felt that public health campaigns, such as Stoptober should not endorse using e-cigarettes as a way to give up smoking tobacco, while 29% (n=149) thought e-cigarettes should be endorsed in campaigns.

## Differences in e-cigarettes practice between clinicians

GPs and practice nurses were significantly more likely to say that they recommended e-cigarettes to cancer patients than the other clinicians included in the study (see table 2).

GPs and practice nurses also rated their knowledge about e-cigarettes higher than did specialist cancer care clinicians, and were also more likely to report having sufficient time to discuss smoking with patients and rated their role in helping patients cut down smoking as more important.

Practice nurses engaged in significantly more behaviours (eg, ascertaining smoking status, advising patients, supporting e-cigarette use in patients) related to smoking cessation with patients than all the other groups. GPs engaged in significantly more behaviours than the other clinicians, but fewer than practice nurses. Practice nurses were significantly more likely to believe in the effectiveness of e-cigarettes in helping cancer patients stop smoking compared with other clinicians.

## Logistic regression model

Table 3 presents a logistic regression model including all COM-B factors and controlling for clinician type. In this model, not recommending e-cigarettes is associated with a lack of knowledge regarding smoking cessation (OR 1.56, 95% CI 1.01 to 2.44) and e-cigarettes (OR 1.64, 95% CI 1.06 to 2.55). Additionally, greater engagement with patients regarding smoking cessation (OR 2.12, 95% CI 1.12 to 4.03), belief in the effectiveness of e-cigarettes (OR 2.36, 95% CI 1.61 to 3.47), belief in sufficient evidence on e-cigarettes (OR 2.08, 95% CI 1.10 to 4.00) and the social opportunity factor of how comfortable they felt discussing e-cigarettes with patients (OR 1.57, 95% CI 1.04 to 2.36) all significantly predicted recommending e-cigarettes to cancer patients. Specifically, those who reported higher levels of engagement around smoking cessation, and those who were comfortable discussing e-cigarettes were more likely to recommend them. However, those who felt the evidence base was lacking were less likely to recommend them.

## DISCUSSION

The findings from this study suggest relatively low levels of clinician support around recommending e-cigarettes to patients with cancer. Despite a growing evidence base to support use and popularity of e-cigarettes as a smoking cessation aid in the UK, nearly one-third of clinicians do not recommend e-cigarettes to patients with cancer who smoke. Not recommending e-cigarettes is associated with a lack of knowledge regarding smoking cessation and e-cigarettes, lack of engagement in smoking cessation practices with patients that smoke, low belief in effectiveness of e-cigarettes, low belief in evidence around e-cigarettes, and not feeling comfortable discussing e-cigarettes with their patients.

In line with the results of this study, a survey of 124 members of The British Thoracic Oncology Group in April 2015 showed that 93% of clinicians agreed that they needed more information and guidance on e-cigarettes

**Table 2**  COM-B measures by HP—mean and SD

| | Total | General practitioner | Practice nurse | Cancer surgeon | Oncologist | Cancer nurse specialist | Test statistic, P value |
|---|---|---|---|---|---|---|---|
| No of participants | 506 | 103 | 102 | 100 | 102 | 99 | |
| Recommend e-cigarettes always/often, N (%) | 97 (19.2) | 24 (23.3) | 36 (35.3) | 9 (9) | 16 (15.7) | 12 (12.1) | $\chi^2$=28.90, <0.001 |
| Capability | | | | | | | |
| Know status | 4.72 (0.59) | 4.75 (0.48) | 4.78 (0.46) | 4.81 (0.46) | 4.75 (0.60) | 4.52 (0.81) | F=4.17, 0.002* |
| General smoking knowledge | 3.40 (1.03) | 3.80 (0.86) | 3.75 (1.08) | 3.10 (1.00) | 3.17 (1.03) | 3.19 (0.95) | F=12.09, <0.001* |
| E-cig knowledge | 2.34 (.99) | 2.40 (0.89) | 2.45 (1.07) | 2.21 (0.94) | 2.54 (1.05) | 2.11 (0.95) | F=3.32, 0.011 |
| Opportunity | | | | | | | |
| Time | 2.60 (1.03) | 2.46 (0.95) | 2.87 (.94) | 2.39 (1.00) | 2.29 (1.06) | 3.02 (1.00) | F=10.47, <0.001* |
| Patient relationships | 2.51 (.68) | 2.45 (0.72) | 2.52 (0.66) | 2.59 (0.64) | 2.49 (0.67) | 2.52 (0.72) | F=0.59, 0.668 |
| Social norms | 3.00 (0.72) | 2.98 (0.86) | 3.18 (0.75) | 2.85 (0.66) | 3.08 (0.65) | 2.91 (0.64) | F=3.36, 0.010 |
| Motivation | | | | | | | |
| Engagement with patients who smoke | 2.77 (0.61) | 2.90 (0.55) | 3.21 (0.55) | 2.60 (0.54) | 2.55 (0.58) | 2.59 (0.55) | F=26.13, <0.001* |
| Effectiveness in helping cancer pts | 2.83 (1.50) | 2.86 (1.48) | 3.39 (1.43) | 2.49 (1.54) | 2.81 (1.45) | 2.57 (1.44) | F=5.89, <0.001* |
| Importance of HP role | 4.35 (0.76) | 4.48 (0.62) | 4.60 (0.60) | 4.33 (0.83) | 4.20 (0.81) | 4.15 (0.84) | F=6.40, <0.001* |
| Lack of evidence N (%) | 242 (47.8) | 57 (55.3) | 51 (50) | 44 (44) | 41 (40.2) | 50 (50.5) | $\chi^2$=5.60, 0.231 |
| Attitudes | 3.22 (0.61) | 3.23 (0.64) | 3.39 (0.62) | 3.08 (0.61) | 3.26 (0.53) | 3.13 (0.59) | F=0.4.04, 0.003 |
| Harm | 3.16 (0.68) | 3.25 (0.63) | 3.15 (0.67) | 3.18 (0.66) | 3.22 (0.71) | 3.01 (0.71) | F=1.32, 0.263 |
| Better than smoking | 3.86 (0.96) | 4.00 (1.00) | 3.84 (1.00) | 3.82 (.81) | 4.07 (0.86) | 3.56 (1.05) | F=4.39, 0.002* |
| Comfortable discussing smoking in general | 4.40 (0.76) | 4.45 (0.75) | 4.44 (0.71) | 4.49 (0.72) | 4.39 (0.81) | 4.24 (0.80) | F=1.59, 0.175 |
| Comfortable discussing e-cigs | 2.91 (1.16) | 3.09 (1.13) | 3.04 (1.23) | 2.65 (1.17) | 3.00 (1.11) | 2.76 (1.12) | F=2.79, 0.026 |

*Significant for F tests when multiple comparisons taken into consideration, accepted p value corrected to 0.003.
COM-B, capability, opportunity and motivation to perform a behaviour.

to advise patients. Clinicians lacked confidence to advise patients with lung cancer to use e-cigarettes.[47] Our findings indicate that, 3 years after this study, clinicians continue to lack knowledge and confidence in recommending e-cigarettes to patients with cancer despite a UK public health policy to support e-cigarettes. Clinicians from various specialties have reported a need for training and local guidance around e-cigarettes in line with the national public health policy.[43 50] Interventions are needed to target the reported behavioural factors associated with clinicians' reluctance to recommend e-cigarettes.

This study highlights that the COM-B components of psychological capability, social opportunity and automatic and reflective motivation were all associated with clinicians' beliefs and behaviours around recommending e-cigarettes. Improving knowledge through accessible training on e-cigarettes alongside local adoption of public health policies around e-cigarettes throughout the NHS may support clinicians to feel more confident and comfortable in recommending them to

patients. The National Institute for Health and Care Excellence in the UK advise that clinicians have an informed discussion with patients on use of e-cigarettes to stop smoking.[51]

A recent study reports that practitioners suggested the development of decision aids around e-cigarettes, such as a leaflet, booklet or online resource to use during consultations with patients.[52] This would aid a more 'neutral' decision around use of e-cigarettes and potentially improve confidence around discussions on e-cigarettes.[53] Furthermore, engagement from local clinical commissioning groups and clinical leads alongside accessible training may support clinicians in providing advice on e-cigarettes to patients with cancer.

To date, there is no medicinally licensed EC in the UK, or anywhere else in the world, which is possibly presenting challenges for clinicians wishing to demonstrate e-cigarettes or recommend for inpatient use. In this study, clinicians were divided over this decision, with two-fifths of clinicians not supportive, and nearly one-third supportive of licensing e-cigarettes for medicinal uses. Clinicians

**Table 3** Results of full binary logistic regression model exploring all factors relating to capability, opportunity and motivation to recommend e-cigarettes to patients with cancer

| | B | Wald (df=1) | P value | 95% CI for OR | | |
| --- | --- | --- | --- | --- | --- | --- |
| | | | | Lower | OR | Upper |
| Capability | | | | | | |
| Know status | 0.758 | 3.320 | 0.068 | 0.944 | 2.134 | 4.821 |
| General smoking knowledge | −0.440 | 3.866 | 0.049 | 1.010 | 1.563 | 2.439 |
| E-cig knowledge | 0.495 | 4.878 | 0.027 | 1.057 | 1.641 | 2.546 |
| Opportunity | | | | | | |
| Time | 0.083 | 0.237 | 0.626 | 0.778 | 1.087 | 1.517 |
| Patient relationships | −0.092 | 0.128 | 0.720 | 0.551 | 0.912 | 1.510 |
| Social norms | −0.047 | 0.028 | 0.867 | 0.549 | 0.954 | 1.658 |
| Motivation | | | | | | |
| Engagement with patients who smoke | 0.752 | 5.308 | 0.021 | 1.119 | 2.122 | 4.025 |
| Effectiveness in helping cancer pts | 0.858 | 19.09 | 0.000 | 1.605 | 2.359 | 3.466 |
| Importance of HP role | 0.259 | 0.834 | 0.361 | 0.743 | 1.295 | 2.257 |
| Sufficient evidence | −0.745 | 4.963 | 0.026 | 1.099 | 2.083 | 4.000 |
| Attitudes | 0.370 | 1.038 | 0.308 | 0.711 | 1.447 | 2.947 |
| Harm | 0.023 | 0.006 | 0.941 | 0.556 | 1.023 | 1.885 |
| Better than smoking | −0.009 | 0.001 | 0.972 | 0.593 | 0.991 | 1.655 |
| Comfortable discussing smoking in general | −0.402 | 2.180 | 0.140 | 0.392 | 0.669 | 1.141 |
| Comfortable discussing e-cigs | 0.448 | 4.594 | 0.032 | 1.039 | 1.565 | 2.356 |
| Health professional (HP) type | | | | | | |
| | | 5.385 | 0.250 | | | |
| GP | 0.276 | 0.256 | 0.613 | 0.453 | 1.318 | 3.839 |
| Practice nurse | 0.281 | 0.294 | 0.588 | 0.479 | 1.324 | 3.659 |
| Cancer surgeon | −0.805 | 1.791 | 0.181 | 0.138 | 0.447 | 1.453 |
| Oncologist | −0.391 | 0.514 | 0.473 | 0.232 | 0.676 | 1.971 |
| Constant | −11.416 | 20.331 | 0.000 | | 0.000 | |

Reference categories—for HP type=cancer nurse specialist, for sufficient evidence=yes, 2.08, 95% CI 1.10 to 4.00.
GP, general practitioner.

may be more comfortable recommending e-cigarettes if they were available on prescription.

Debate is also needed on how e-cigarettes could be integrated into smoking cessation practices delivered by clinicians to patients with cancer. Smoking cessation practices are already well developed in other disease groups such as coronary heart disease.[54] It is, therefore, timely to examine the role of a smoking cessation service within the cancer pathway, including the role of e-cigarettes can play in helping cancer patients to quit smoking long term after a diagnosis.[55] While attendance at a cancer clinic provides an opportunity for clinicians to provide smoking cessation support to those who smoke, this is not currently routinely offered. Patients with cancer have highlighted the need for smoking cessation support, and have reported difficulty in attending external smoking cessations services in addition to all their other clinic appointments.[12] Smoking cessation advice has historically been the role of primary care or community smoking cessation services, and this is reflected in the greater knowledge, confidence and more positive attitude towards e-cigarettes among practice nurses and GPs than among cancer specialists reported in this study. However, lack of funding has seen a decline in the smoking cessation services and alternatives are needed. Clinicians worry that discussing smoking may damage the relationship with the patient, which is essential for the often onerous treatment needed for cancer.[53] Clinicians also seem to believe that patients cannot stop smoking.[56]

### Strengths and limitations
To our knowledge, this study is the first to examine a broad range of clinicians' behaviours and beliefs around the use of e-cigarettes in patients across different cancer groups. The COM-B model has previously been used to develop smoking cessation interventions.[57] In this study, it has enabled the identification of factors, which could be used to improve clinicians' recommendation of e-cigarettes.

The recruitment procedure for this survey used an existing network of electronically active clinicians from the research arm of doctors.net.com. This method has the advantage of speed and guaranteed response which is beneficial considering that surveys with busy clinicians have commonly suffered from poor response rates. However, this sample may not be representative of the population of clinicians in the UK. Quota sampling ensured diversity in our sample, which included clinicians who worked with patients who had a wide range of cancer diagnoses, and not just those with cancers directly associated with smoking. However, in quota sampling, the sample has not been chosen using random selection

The potential for response bias should be considered; for example, those with a greater interest in smoking cessation for cancer patients may have been more likely to respond, and the incentive may have encouraged participation, whereas clinicians who are smokers may have been under-represented. In addition, findings rely on self-report.

## CONCLUSIONS

Despite the evidence that e-cigarettes help smokers quit smoking, and the positive public health stance towards e-cigarettes in the UK, clinicians remain cautious about recommending e-cigarettes to cancer survivors who continue to smoke. Clinicians require training and support on how to integrate e cigarettes in smoking cessation advice for patients with cancer and adoption of the UK evidence based guidance at regional and local level is needed.

**Author affiliations**
[1]Health & Life Sciences, Oxford Brookes University, Oxford, UK
[2]Department of Psychology, Social Workand Public Health Oxford Brookes University, Oxford, UK
[3]Nuffield Department of Primary Care Health Sciences, University of Oxford, Oxford, UK
[4]Department of Surgery & Cancer, Faculty of Medicine, Imperial College Healthcare NHS Trust, London, UK
[5]Cancer Centre, Oxford University Hospitals NHS Foundation Trust, Oxford, UK
[6]Molecular, Genetic and Population Health Sciences, University of Edinburgh, Edinburgh, Scotland
[7]Vaper Representative, London, UK

**Acknowledgements** This work was funded and supported by Cancer Research UK. We thank the clinicians who participated in this online survey, and Mede Connect Health care Insight, the research arm of Doctors. Net.UK for recruiting, sending and collating results for the online survey. Summary results have been presented at National Cancer Research Institute (NCRI) conference Glasgow November 2018, and International Psycho-Oncology Society (IPOS) conference in Banff, Canada, September 2019E-cigarettes as a Smoking Cessation aid in Cancer Patients: Clinicians knowledge, attitude and current practice. NCRI 2018 abstract 2171E-cigarettes as a Smoking Cessation aid in Cancer Patients: Clinicians knowledge, attitude and current practice. IPOS 2019 abstract 751. Journal of Psychosocial Oncology – Research and Practice Sept 2019, vol 1: suppl 1 1S.

**Contributors** JB, ELD, EW, PA, FM, DF and MW conceived and designed the study. Additionally, LS, BDN, SDSM and SJ commented on the final funding application before submission. All authors were involved in the development of the protocol. FM, JB and ELD developed the initial questionnaire, with e-cigarette knowledge input from PA, SJ and LS, primary health care input from PA & BDN, nursing input from MW and surgeon and oncology input from SDSM. ELD and FM conducted the statistical analysis, which was reviewed by DF. JB and FM wrote the initial manuscript. EW, PA, ELD, DF and MW critically reviewed the initial manuscript, while LS, BDN, SDSM and SJ reviewed and commented on the second version of the manuscript. All authors read and approved the final manuscript.

**Funding** Cancer Research UK Ref: A24355 JB. PA is an NIHR senior investigator and funded by the NIHR Oxford Biomedical Research Centre and the Applied Research Centre.

**Competing interests** None declared.

**Patient and public involvement** Patients and/or the public were involved in the design, or conduct, or reporting, or dissemination plans of this research. Refer to the Methods section for further details.

**Patient consent for publication** Not required.

**Ethics approval** This study complies with ethical standards. The study obtained ethics approval to conduct the study from Oxford Brookes University Ethics Committee (2017 44 Brett) and was performed in accordance with the Declaration of Helsinki.

**Provenance and peer review** Not commissioned; externally peer reviewed.

**Data availability statement** Data are available in a public, open access repository. Data are available on reasonable request. Data will be deposited in the university repository, where reasonable requests for access can be submitted. Data access can be requested from the corresponding author. The data will be kept for a minimum of ten years following publication.

**ORCID iDs**
Jo Brett http://orcid.org/0000-0002-5116-4238
Paul Aveyard http://orcid.org/0000-0002-1802-4217
Mary Wells http://orcid.org/0000-0001-5789-2773
David Foxcroft http://orcid.org/0000-0001-9752-7527
Brian Nicholson http://orcid.org/0000-0003-0661-7362
Eila Watson http://orcid.org/0000-0002-3592-1315

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
