## [Reviewer comments · BMJ Open]

ARTICLE DETAILS

TITLE (PROVISIONAL)	Electronic cigarettes as a smoking cessation aid for cancer patients: beliefs and behaviours of clinicians in the UK
AUTHORS	Brett, Jo; Davies, EL; Matley, Fiona; Aveyard, Paul; Wells, Mary; Foxcroft, David; Nicholson, Brian; De Silva Minor, Shiroma; Sinclair, Lesley; Jakes, Sarah; Watson, Eila

VERSION 1 – REVIEW

REVIEWER	Wojciech S. Zgliczyński School of Public Health, Centre of Postgraduate Medical Education, Warsaw, Poland
REVIEW RETURNED	09-Mar-2020

GENERAL COMMENTS	This is a well-prepared manuscript presenting findings from a cross-sectional survey concerning a significant public health issue. The study is novel and the findings are new and relevant for tobacco control researchers. The authors' aimed to "explore clinicians' beliefs and behaviours around recommending e-cigarettes as a smoking cessation aid for cancer patients". The main comment is that the manuscript should be more balanced. PHE is a sole organization that considers e-cigarettes as a safer alternative compared to combustible cigarettes and promotes "switching" from smoking to vaping. It should be underlined within the manuscript, that these results can't be generalized to other European countries, because the British approach to e-cigarette is an exception in Europe. Some minor remarks would require corrections to improve the quality of the paper: 1) Please try to avoid writing new sentences starting from numbers eg. X% of clinicians.... etc. Please consider rewriting these sentences.2) Page 4 lines 48-55. These sentences are too far-reaching. The Authors provided only 1 reference [25] to support the statement "recent evidence suggests that e-cigarettes are more effective for smoking cessation than nicotine-replacement therapy when both products are accompanied by behavioural support.". Please provide sufficient scientific evidences to support these theses or rewrite this sentence (more balanced statement).3) Please provide the number of physicians enrolled in this study and response rate. Page 5, lines 22-254) Sampling method should be described in more detail.5) Please provide more data about the pilot (5 GPs, 5 cancer patients) - measured outcomes? what was the impact of the pilot on the final questionnaire?6) Page 6, line 54 please provide a study questionnaire of an active link.
---

	7) Page 8, line 13 - please provide more information about financial rewards? How about the founder? Did the study sponsor affect the questions? Is there a bias risk due to financial reward? 8) Table 1 - please consider change "Mean year qualified as a health professional" to "years of professional experience". It would be more informative. 9) Page 16 - lines 47-58 - this paragraph must be rewritten. "Opportunity to recommend e-cigarettes could be improved by providing a more positive strategy around e-cigarettes. An ongoing ban of e-cigarettes in many NHS organizations, or permitted use only in dedicated smoking areas not only limits clinicians from demonstrating e-cigarettes but puts vapers at risk of relapsing. Allowing vaping in appropriate parts of the NHS may improve compliance with public health policy around e-cigarettes." This is the individual opinion of the authors without scientific evidence. There is a wide scientific debate on whether e-cigarettes can be recommended as a safer alternative. All tobacco control policies (including WHO FCTC) recommends limiting access to tobacco and vaping products. Medical facilities should be tobacco/nicotine-free. 10) Limitations section should be more extensive, including sampling method and impact of financial reward for participants.
--	--

REVIEWER	Andrew Nickels Park Nicollet Health Services, United States
REVIEW RETURNED	13-Apr-2020

GENERAL COMMENTS	Response rate needs to included. Write up seems bias toward the use of e-cigarettes. I do not know of studies in cancer patient using e-cigs for smoking cessation. My understanding of UK policy suggests a balanced approach toward e-cig use in cessation attempts, not necessarily a pro-e cig stance. Current manuscript concludes e-cig use should be taken up more by providers treating patients but it might be interpreted as providers are not confident in e-cigs or recommendations enough to adopt them.
--

REVIEWER	Jarosław Pinkas School of public health, Centre of postgraduate medical education, Warsaw, Poland
REVIEW RETURNED	23-Apr-2020

GENERAL COMMENTS	This is a well-written manuscript presenting findings from a cross-sectional survey concerning a significant public health issue. There are some minor remarks that would require corrections to improve the quality of the paper: 1) Introduction section (page 4, lines 48-51) - please consider adding 1-2 sentences about the global debate around the e-cigarettes and its potential role as a smoking cessation tool. The reviewer is aware that the PHE recommends e-cigs as a smoking cessation tool, however, this issue raises debate around Europe. Please consider adding 2 sentences to broaden the international context of this paper. 2) Page 8 line 2 -> please describe how physicians who care for cancer patients in the UK were defined? self-declared? or official physicians' registry?
---

	3) Financial reward for the participants may impact the potential response bias, please consider adding this in the limitations section. 4) Results - information about response rate should be placed (if available) 5) Table 1 - please consider "years of professional experience" rather than "Mean year qualified as a health professional" - it would be more informative 6) Table 3 - please unify a number of decimal places 7) Discussion - please briefly compare the results to other European countries and the US - it would be interesting to compare it, to the countries with a different approach to e-cigs and vaping products The main limitation of this study is the sampling method, which may lead to response bias and limits the possibility of generalizing results to other medical professional groups
--	--

VERSION 1 – AUTHOR RESPONSE

Reviewer(s)' Comments to Author:

Reviewer: 1

Reviewer Name: Wojciech S. Zgliczyński

Institution and Country: School of Public Health, Centre of Postgraduate Medical Education, Warsaw, Poland

Please state any competing interests or state 'None declared': None Declared

Please leave your comments for the authors below

This is a well-prepared manuscript presenting findings from a cross-sectional survey concerning a significant public health issue. The study is novel and the findings are new and relevant for tobacco control researchers.

Author response: Thank-you

The authors' aimed to "explore clinicians' beliefs and behaviours around recommending e-cigarettes as a smoking cessation aid for cancer patients".

The main comment is that the manuscript should be more balanced. PHE is a sole organization that considers e-cigarettes as a safer alternative compared to combustible cigarettes and promotes "switching" from smoking to vaping. It should be underlined within the manuscript, that these results can't be generalized to other European countries, because the British approach to e-cigarette is an exception in Europe.

Author response: We agree that the manuscript should be more balanced and we have added several paragraphs to the introduction that discuss the different regulations for e-cigarettes in different countries (see below). However, 66 countries in the world sell e-cigarettes, and while the policy in the UK remains more positive than most around the use of e-cigarettes as a smoking cessation aid, a growing body of evidence suggests that e-cigarettes are substantially less harmful than smoking tobacco cigarettes and that potential shifts towards a risk reduction policy in response to this evidence may make the results of this study generalizable outside of the UK.

A risk reduction policy is particularly important in people who have been diagnosed with cancer, but despite this continue to smoke and therefore increase their risk of premature death as a result.

We believe that as the evidence base around e-cigarettes continues to grow, the debate around both the positive and negative evidence is important internationally. Journals continue to publish studies that evaluate the use of e-cigarettes as a smoking cessation aid, and this should be considered and inform policy in the future.

Changes to text in introduction

In recent years e-cigarettes have grown in popularity as a cessation aid among smokers worldwide, but the e-cigarette regulatory environment in each country varies considerably due to policy makers' reactions to a rapidly developing evidence base.^{19,20} Often this is influenced by strategic positions on tobacco control e.g. harm prevention vs. harm reduction,²¹ and whether e-cigarettes are classified as a tobacco, medicinal or consumer product.¹⁹ The main debates focus on trying to achieve a balance between the risks and potential of e-cigarettes.²¹

Support for the use of e-cigarettes for smoking cessation is endorsed by several health-related organisations in the UK including Public Health England,²² Cancer Research UK,²³ the National Health Service²⁴ and the National Centre for Smoking Cessation and Training.²⁵ Public Health Scotland,²⁶ Public Health Wales,²⁷ the British Medical Association,²⁸ the Royal College of General Practitioners²⁹ and the Royal College of Physicians³⁰ also acknowledge that e-cigarettes are considered less harmful than smoking tobacco cigarettes or that some people may find using e-cigarettes useful for stopping or reducing smoking.

In the UK and other European countries, the manufacture, presentation and sales of e-cigarettes are regulated by the Tobacco Products Directive (2014/40/EU).^{31, 32} The regulation prohibits sales of e-cigarettes to people under 18 years, most forms of advertising and places restrictions on the type and quality of ingredients. In France there are further restrictions on the use of e-cigarettes in public places.³³ The regulatory environment differs substantially in other parts of the world.

In the United States, for instance, the Centres for Disease Control and Prevention (CDC) advice is not unlike many UK public health bodies in stating that "if (adults) choose to use e-cigarettes as an alternative to cigarettes, they should completely switch from cigarettes to e-cigarettes and not partake in an extended period of dual use of both products that delays quitting smoking completely".³⁴ The World Health Organisation position is that there is insufficient evidence concerning comparisons with combustible cigarettes or efficacy for smoking cessation, and that the use of e-cigarettes is harmful.³⁵ In October, 2018, 72 experts with no connections to the tobacco industry wrote to the WHO Director-General to argue that WHO should embrace innovation and more actively include tobacco harm reduction in its strategy to tackle the burden of smoking-related disease.

Whilst the need for further evidence around impacts of e-cigarettes on cessation and long-term harms is widely acknowledged, many health-related organisations also recognise that e-cigarettes may have a potential role in smoking cessation and could help people that may otherwise continue to smoke. A risk reduction policy to encourage smoking cessation is particularly important in people who have been diagnosed with cancer who continue smoking and therefore increase their risk of recurrence, other co-morbidities and premature death as a result. 4-11

Some minor remarks would require corrections to improve the quality of the paper:

1) Please try to avoid writing new sentences starting from numbers e.g. X% of clinicians.... etc. Please consider rewriting these sentences.

Author response: This is a good point. We have now re-written the start of these sentences.

2) Page 4 lines 48-55. These sentences are too far-reaching. The Authors provided only 1 reference [25] to support the statement “recent evidence suggests that e-cigarettes are more effective for smoking cessation than nicotine-replacement therapy when both products are accompanied by behavioural support.”. Please provide sufficient scientific evidences to support these theses or rewrite this sentence (more balanced statement).

We have now re-written this sentence to state that ‘a recent study reports that e-cigarettes are more effective for smoking cessation than nicotine replacement therapy when both products are accompanied by behavioural support’.

3) Please provide the number of physicians enrolled in this study and response rate. Page 5, lines 22-25

Author response: Quota sampling was used to stratify the sample by type of clinician and by the seven National Health Service (NHS) UK regions. When the quota of responses from each type of clinician or from each UK NHS region was reached, the survey was closed for that clinician group or region. The online survey went out to all relevant health professionals on the research arm of doctors.net.uk. We have added the number of clinician type on their database to the methods section. As this was an anonymous online survey, where participation was closed when the quota was reached, it is not possible to calculate the response rate.

4) Sampling method should be described in more detail.

Author response:

Quota sampling was used to stratify the sample by type of clinician and by the seven National Health Service (NHS) UK regions. Quota sampling method was selected as a sampling method to gather diverse data from the relevant clinician population. When the quota of responses from each type of clinician or from each UK NHS region was reached, the survey was closed for that clinician group or region.

We have added the following:

Sampling was restricted to currently practising clinicians, was stratified by NHS region, and was conducted on a ‘first come, first served’ basis; the target number of responses was 100 for each of the clinician types. When the quota of responses from each type of clinician or from each UK NHS region was reached, the survey was closed for that clinician group or region.

5) Please provide more data about the pilot (5 GPs, 5 cancer patients) - measured outcomes? what was the impact of the pilot on the final questionnaire?

Author response: We appreciate that the use of the word pilot was incorrect. The face validity of the survey was evaluated in 5 GPs and 5 cancer clinicians to evaluate whether the survey seemed appropriate and the questions were understood. Minor changes to wording were made to ensure the right interpretation of the questions. We have now re-worded this paragraph as:

The face validity of the survey was evaluated in 5 general practitioners (GPs) and 5 cancer clinicians to evaluate whether the survey was appropriate and the questions were understood. This resulted in modification of the wording of some questions.

6) Page 6, line 54 please provide a study questionnaire of an active link.

The questionnaire has been uploaded to the journal webpage for an active link to be provided

7) Page 8, line 13 - please provide more information about financial rewards? How about the founder? Did the study sponsor affect the questions? Is there a bias risk due to financial reward?

The financial incentive was a token incentive to cover the time taken to complete the survey, as the survey was completed outside of NHS hours. The incentive did not completely cover the cost of their time, but was provided as a token of appreciation for taking the time to complete the questionnaire. A survey that is not randomly administered will result in very low response rates among busy clinicians. As such, it is highly likely to attract people interested in the topic. Paying people who are completing the survey as a means to earn money means that participation is less likely to be biased because people are motivated by incentive rather than topic-related views and thus may be more representative.

Incentives can increase the perception of value, trust, reciprocity, and appreciation on the part of the respondent (Cho, Johnson & Vangeest 2013; Davern 2013). However, we also appreciate that incentives can cause participation bias and this has been added to the limitations in the Strengths and Limitations section.

The funder is the biggest cancer charity in the UK, Cancer Research UK, and were not involved in the development of the questions.

Cho YI, Johnson TP, Vangeest JB. Enhancing surveys of health care professionals: a meta-analysis of techniques to improve response. *Eval Health Prof* 2013 Sep;36(3):382-407.

Davern M. Nonresponse rates are a problematic indicator of nonresponse bias in survey research. *Health Serv Res* 2013 Jun;48(3):905-912

8) Table 1 - please consider change "Mean year qualified as a health professional" to "years of professional experience". It would be more informative.

This change has now been made

9) Page 16 - lines 47-58 - this paragraph must be rewritten. "Opportunity to recommend e-cigarettes could be improved by providing a more positive strategy around e-cigarettes. An ongoing ban of e-cigarettes in many NHS organizations, or permitted use only in dedicated smoking areas not only limits clinicians from demonstrating e-cigarettes but puts vapers at risk of relapsing. Allowing vaping in appropriate parts of the NHS may improve compliance with public health policy around e-cigarettes." This is the individual opinion of the authors without scientific evidence. There is a wide scientific debate on whether e-cigarettes can be recommended as a safer alternative. All tobacco control policies (including WHO FCTC) recommends limiting access to tobacco and vaping products. Medical facilities should be tobacco/nicotine-free.

Author response: We have now deleted this paragraph as we agree it is an opinion. There is debate within the UK whether there should be a separate zone for vapers away from smokers at NHS sites that provide smoking zones outside of their facilities, and a wider debate for all companies in the UK. As UK policy is to support use of e-cigarettes as a smoking cessation aid, asking vapers to share the same zones as smokers may tempt them back to smoking and will result in them inhaling non-filtered smoke. A recent editorial in the *Lancet* debates this issue alongside other issues around e cigarettes:

Beaglehole R, Bates C, Youdan B, Bonita R. Nicotine without smoke: fighting the tobacco epidemic with harm reduction. *Lancet* Vol 394 August 31, 2019.

10) Limitations section should be more extensive, including sampling method and impact of financial reward for participants.

Author Response: Limitations of sampling and incentives are now reported in the limitations section

Reviewer: 2

Reviewer Name: Andrew Nickels

Institution and Country: Park Nicollet Health Services, United States

Please state any competing interests or state 'None declared': None Declared

Please leave your comments for the authors below

Response rate needs to included.

Author response: The online survey was anonymous and was sent to all the relevant UK health professionals registered with doctors.net.uk who had agreed to be involved in research through their research arm medeconnect (M3 Europe). Quota sampling was used where the survey was closed to clinician groups when a sample size of 100 was reached. It is therefore not possible to calculate a response rate. One of the cancer nurse specialists was excluded after the survey was closed because she worked with children. We have added the number of clinician type on the M3 database.

Write up seems bias toward the use of e-cigarettes. I do not know of studies in cancer patient using e-cigs for smoking cessation. My understanding of UK policy suggests a balanced approach toward e-cig use in cessation attempts, not necessarily a pro-e cig stance.

Author response: We have tried to address the bias towards use of e-cigarettes in the background section. Please see additional sections added below:

In recent years e-cigarettes have grown in popularity as a cessation aid among smokers worldwide, but the e-cigarette regulatory environment in each country varies considerably due to policy makers' reactions to a rapidly developing evidence base.^{19,20} Often this is influenced by strategic positions on tobacco control e.g. harm prevention vs. harm reduction,²¹ and whether e-cigarettes are classified as a tobacco, medicinal or consumer product.¹⁹ The main debates focus on trying to achieve a balance between the risks and potential of e-cigarettes.²¹

Support for the use of e-cigarettes for smoking cessation is endorsed by several health-related organisations in the UK including Public Health England,²² Cancer Research UK,²³ the National Health Service²⁴ and the National Centre for Smoking Cessation and Training.²⁵ Public Health Scotland,²⁶ Public Health Wales,²⁷ the British Medical Association,²⁸ the Royal College of General Practitioners²⁹ and the Royal College of Physicians³⁰ also acknowledge that e-cigarettes are considered less harmful than smoking tobacco cigarettes or that some people may find using e-cigarettes useful for stopping or reducing smoking.

In the UK and other European countries, the manufacture, presentation and sales of e-cigarettes are regulated by the Tobacco Products Directive (2014/40/EU).^{31, 32} The regulation prohibits sales of e-cigarettes to people under 18 years, most forms of advertising and places restrictions on the type and

quality of ingredients. In France there are further restrictions on the use of e-cigarettes in public places.³³ The regulatory environment differs substantially in other parts of the world.

In the United States, for instance, the Centres for Disease Control and Prevention (CDC) advice is not unlike many UK public health bodies in stating that “if (adults) choose to use e-cigarettes as an alternative to cigarettes, they should completely switch from cigarettes to e-cigarettes and not partake in an extended period of dual use of both products that delays quitting smoking completely”.³⁴ The World Health Organisation position is that there is insufficient evidence concerning comparisons with combustible cigarettes or efficacy for smoking cessation, and that the use of e-cigarettes is harmful.³⁵, although In October, 2018, 72 experts with no connections to the tobacco industry wrote to the WHO Director-General to argue that WHO should embrace innovation and more actively include tobacco harm reduction in its strategy to tackle the burden of smoking-related disease.

A risk reduction policy is particularly important in people who have been diagnosed with cancer, but despite this continue to smoke and risk premature death as a result. The following papers have explored the use of e-cigarettes in cancer patients. We have listed some relevant publications below:

Antwi 2019

Associations between e-cigarette and combustible cigarette use among U.S. cancer survivors: implications for research and practice

<https://pubmed-ncbi-nlm-nih-gov.oxfordbrookes.idm.oclc.org/30955182/>

Borderud 2014

Electronic cigarette use among patients with cancer: characteristics of electronic cigarette users and their smoking cessation outcomes

<https://pubmed-ncbi-nlm-nih-gov.oxfordbrookes.idm.oclc.org/25252116/>

Erratum: <https://pubmed-ncbi-nlm-nih-gov.oxfordbrookes.idm.oclc.org/25855820/>

Reply: <https://pubmed-ncbi-nlm-nih-gov.oxfordbrookes.idm.oclc.org/25740086/>

Buczek 2018

Electronic Cigarette Awareness, Use, and Perceptions among Cancer Patients

<https://pubmed-ncbi-nlm-nih-gov.oxfordbrookes.idm.oclc.org/30480216/>

Correa 2018

Electronic cigarette use among patients with cancer: Reasons for use, beliefs, and patient-provider communication

<https://pubmed-ncbi-nlm-nih-gov.oxfordbrookes.idm.oclc.org/29671928/>

Kalkhoran 2018

Electronic cigarette use patterns and reasons for use among smokers recently diagnosed with cancer

<https://pubmed-ncbi-nlm-nih-gov.oxfordbrookes.idm.oclc.org/29905013/>

Fahey 2019

Prevalence and correlates of dual tobacco use in cancer survivors

<https://pubmed-ncbi-nlm-nih-gov.oxfordbrookes.idm.oclc.org/30671688/>

McQueen 2016

Smoking Cessation and Electronic Cigarette Use among Head and Neck Cancer Patients

<https://pubmed-ncbi-nlm-nih-gov.oxfordbrookes.idm.oclc.org/31838445/>

Salloum 2016

Use of Electronic Cigarettes Among Cancer Survivors in the U.S.
<https://pubmed-ncbi-nlm-nih-gov.oxfordbrookes.idm.oclc.org/27242079/>

Salloum 2019
Tobacco and E-cigarette use among cancer survivors in the United States
<https://pubmed-ncbi-nlm-nih-gov.oxfordbrookes.idm.oclc.org/31815948/>

Symes 2019
Dual cigarette and e-cigarette use in cancer survivors: an analysis using
Population Assessment of Tobacco Health (PATH) data
<https://pubmed-ncbi-nlm-nih-gov.oxfordbrookes.idm.oclc.org/30675695/>

Zeng 2019 (Cochrane Review)
Interventions for smoking cessation in people diagnosed with lung cancer
<https://pubmed-ncbi-nlm-nih-gov.oxfordbrookes.idm.oclc.org/31173336/>

Akinboro 2019
Electronic Cigarette Use among Survivors of Smoking-Related Cancers in the United States
<https://cebp.aacrjournals.org/content/28/12/2087.long>

Protocol:
Begh 2019
Examining the effectiveness of general practitioner and nurse promotion of electronic cigarettes
versus standard care for smoking reduction and abstinence in hardcore smokers with smoking-related
chronic disease: protocol for a randomised controlled trial
<https://pubmed-ncbi-nlm-nih-gov.oxfordbrookes.idm.oclc.org/31779689/>

Screening:
Lucchiari 2020
Benefits of e-cigarettes in smoking reduction and in pulmonary health among chronic smokers
undergoing a lung cancer screening program at 6 months
<https://pubmed-ncbi-nlm-nih-gov.oxfordbrookes.idm.oclc.org/31838445/>

Current manuscript concludes e-cig use should be taken up more by providers treating patients but it
might be interpreted as providers are not confident in e-cigs or recommendations enough to adopt
them.

The conclusion has now been changed to :

Despite the evidence that e-cigarettes help smokers quit smoking, and the positive public health
stance towards e-cigarettes in the UK, clinicians remain cautious about recommending e-cigarettes to
cancer survivors who continue to smoke. Clinicians require training and support on how to integrate e
cigarettes in smoking cessation advice for cancer patients and adoption of the UK evidence based
guidance at regional and local level is needed.

Reviewer: 3
Reviewer Name: Jarosław Pinkas

Institution and Country: School of public health, Centre of postgraduate medical education, Warsaw, Poland

This is a well-written manuscript presenting findings from a cross-sectional survey concerning a significant public health issue.

Author response: Thank-you

There are some minor remarks that would require corrections to improve the quality of the paper:

1) Introduction section (page 4, lines 48-51) - please consider adding 1-2 sentences about the global debate around the e-cigarettes and its potential role as a smoking cessation tool. The reviewer is aware that the PHE recommends e-cigs as a smoking cessation tool, however, this issue raises debate around Europe. Please consider adding 2 sentences to broaden the international context of this paper.

Author response: We have tried to address this comment by adding several paragraphs to the introduction of the paper. Please see paragraphs added below:

In recent years e-cigarettes have grown in popularity as a cessation aid among smokers worldwide, but the e-cigarette regulatory environment in each country varies considerably due to policy makers' reactions to a rapidly developing evidence base.^{19,20} Often this is influenced by strategic positions on tobacco control e.g. harm prevention vs. harm reduction,²¹ and whether e-cigarettes are classified as a tobacco, medicinal or consumer product.¹⁹ The main debates focus on trying to achieve a balance between the risks and potential of e-cigarettes.²¹

Support for the use of e-cigarettes for smoking cessation is endorsed by several health-related organisations in the UK including Public Health England,²² Cancer Research UK,²³ the National Health Service²⁴ and the National Centre for Smoking Cessation and Training.²⁵ Public Health Scotland,²⁶ Public Health Wales,²⁷ the British Medical Association,²⁸ the Royal College of General Practitioners²⁹ and the Royal College of Physicians³⁰ also acknowledge that e-cigarettes are considered less harmful than smoking tobacco cigarettes or that some people may find using e-cigarettes useful for stopping or reducing smoking.

In the UK and other European countries, the manufacture, presentation and sales of e-cigarettes are regulated by the Tobacco Products Directive (2014/40/EU).^{31, 32} The regulation prohibits sales of e-cigarettes to people under 18 years, most forms of advertising and places restrictions on the type and quality of ingredients. In France there are further restrictions on the use of e-cigarettes in public places.³³ The regulatory environment differs substantially in other parts of the world.

In the United States, for instance, the Centres for Disease Control and Prevention (CDC) advice is not unlike many UK public health bodies in stating that "if (adults) choose to use e-cigarettes as an alternative to cigarettes, they should completely switch from cigarettes to e-cigarettes and not partake in an extended period of dual use of both products that delays quitting smoking completely".³⁴ The World Health Organisation position is that there is insufficient evidence concerning comparisons with combustible cigarettes or efficacy for smoking cessation, and that the use of e-cigarettes is harmful.³⁵ In October, 2018, 72 experts with no connections to the tobacco industry wrote to the WHO Director-General to argue that WHO should embrace innovation and more actively include tobacco harm reduction in its strategy to tackle the burden of smoking-related disease.

2) Page 8 line 2 -> please describe how physicians who care for cancer patients in the UK were defined? self-declared? or official physicians' registry?

Author Response: All clinicians who are registered with doctors.net.uk have to provide their clinical registration number during the registration process. The clinicians are therefore officially registered clinicians. We have now added this to the methods section

3) Financial reward for the participants may impact the potential response bias, please consider adding this in the limitations section.

This has now been added to the limitations section

4) Results - information about response rate should be placed (if available)

Author response: The online survey was anonymous and was sent to all the relevant UK health professionals registered with doctors.net.uk who had agreed to be involved in research through their research arm medeconnect (M3 Europe). Quota sampling was used where the survey was closed to clinician groups when a sample size of 100 was reached. It is therefore not possible to calculate a response rate. One of the cancer nurse specialists was excluded after the survey was closed because she worked with children. We have added the number of clinician type on the M3 database.

5) Table 1 - please consider "years of professional experience" rather than "Mean year qualified as a health professional" - it would be more informative

This has now been changed in Table 1

6) Table 3 - please unify a number of decimal places

This has now been changed in Table 3

7) Discussion - please briefly compare the results to other European countries and the US - it would be interesting to compare it, to the countries with a different approach to e-cigs and vaping products

This discussion has been added to the Background section of the paper

The main limitation of this study is the sampling method, which may lead to response bias and limits the possibility of generalizing results to other medical professional groups.

We have now added this issue to the limitations section of the paper

Quota sampling was used to stratify the sample by type of clinician and by the seven National Health Service (NHS) UK regions. Quota sampling method was selected as a sampling method to gather diverse data from the relevant clinician population. When the quota of responses from each type of clinician or from each UK NHS region was reached, the survey was closed for that clinician group or region.

Response rates to clinician surveys are often low due to time limitations and the plethora of surveys that they are asked to get involved in. Incentives to complete the survey may increase the rate of response (Abdulaziz et al, 2014).

Abdulaziz K, Brehaut J, Taljaard M, Emond M, Sirois MJ, Lee JS, Wilding L, Perry J. National survey of physicians to determine the effect of unconditional incentives on response rates of physician postal surveys. *BMJ Open* <http://dx.doi.org/10.1136/bmjopen-2014-007166>

VERSION 2 – REVIEW

REVIEWER	Wojciech S. Zgliczyński School of Public Health, Centre of Postgraduate Medical Education, Warsaw, Poland
REVIEW RETURNED	11-Sep-2020

GENERAL COMMENTS	The Author's responses are satisfactory. All the suggested changes were applied and the final version of the manuscript is balanced.
--

REVIEWER	Jarosław Pinkas School of Public Health, Centre of Postgraduate Medical Education, Warsaw, Poland
REVIEW RETURNED	06-Sep-2020

GENERAL COMMENTS	The authors of this paper addressed the comments of the reviewer comprehensively
--